# Gut Microbiota Regulation and Their Implication in the Development of Neurodegenerative Disease

**DOI:** 10.3390/microorganisms9112281

**Published:** 2021-11-02

**Authors:** Peilin Sun, Lei Su, Hua Zhu, Xue Li, Yaxi Guo, Xiaopeng Du, Ling Zhang, Chuan Qin

**Affiliations:** 1NHC Key Laboratory of Human Disease Comparative Medicine, Institute of Laboratory Animal Sciences, Chinese Academy of Medical Sciences (CAMS), Beijing 100021, China; sunpeilin@cnilas.org (P.S.); sulei@cnilas.org (L.S.); zhuh@cnilas.org (H.Z.); lixue@cnilas.org (X.L.); GYX409@163.com (Y.G.); du_xiaopeng@163.com (X.D.); zhangling@cnilas.org (L.Z.); 2Beijing Engineering Research Center for Experimental Animal Models of Human Critical Diseases, Chinese Academy of Medical Sciences (CAMS), Beijing 100021, China

**Keywords:** gut microbiota, gut–brain axis, neurodegenerative diseases, Alzheimer’s disease, Parkinson’s disease, mechanism, metabolites, therapy, probiotics, fecal microbiota transplantation

## Abstract

In recent years, human gut microbiota have become one of the most promising areas of microorganism research; meanwhile, the inter-relation between the gut microbiota and various human diseases is a primary focus. As is demonstrated by the accumulating evidence, the gastrointestinal tract and central nervous system interact through the gut–brain axis, which includes neuronal, immune-mediated and metabolite-mediated pathways. Additionally, recent progress from both preclinical and clinical studies indicated that gut microbiota play a pivotal role in gut–brain interactions, whereas the imbalance of the gut microbiota composition may be associated with the pathogenesis of neurological diseases (particularly neurodegenerative diseases), the underlying mechanism of which is insufficiently studied. This review aims to highlight the relationship between gut microbiota and neurodegenerative diseases, and to contribute to our understanding of the function of gut microbiota in neurodegeneration, as well as their relevant mechanisms. Furthermore, we also discuss the current application and future prospects of microbiota-associated therapy, including probiotics and fecal microbiota transplantation (FMT), potentially shedding new light on the research of neurodegeneration.

## 1. Introduction

In recent years, human gut microbiota have become one of the most promising areas of microorganism research. To date, a large number of studies reveal that gut microbiota influence a wide range of human pathologies, including inflammatory bowel disease (IBD) [1,2,3]; irritable bowel syndrome (IBS) [4,5]; allergic [6,7,8], neurological [9,10,11], and metabolic diseases [12,13,14]; and psychiatric disorders [15], etc. The latest studies have begun to understand the mechanism by which gut microbiota affect the brain [16,17,18]. Additionally, the inter-communication between gut microbiota and the central nervous system has become increasingly evident.

The gut–brain axis is a two-way functional communication network between the intestine and the brain, which primarily includes neuroendocrine, neural, endocrine and immune signaling pathways [19,20] (Figure 1). The metabolites of intestinal microbiota, such as short-chain fatty acids, gamma aminobutyric acid (GABA), serotonin, kynurenines, norepinephrine, and histamine, etc., regulate a series of cerebral physiological processes in the brain through these pathways [21,22,23,24]. When the composition of the intestinal microbiome falls out of balance (i.e., dysbiosis), signals are sent to the brain that subsequently manifest as low-grade inflammation, heightened oxidative stress, disrupted energy metabolism and increased cellular degeneration [25], contributing to the pathological processes of assorted neurological diseases, particularly neurodegeneration [26,27]. On the other hand, aging in humans is reportedly also related to significant shifts in the composition of gut microbiota, and the loss of microbial diversity was also evident in the aging gut [28,29]. In addition, a striking alteration of microbiota composition was observed in the gastrointestinal tracts of elderly patients suffering from neurodegeneration [29]. Hence, researchers hypothesized that the alteration of human gut microbiota might well be one of the causes, or at least one of the contributing factors, of neurodegenerative diseases. In this review, we summarized the existing evidence regarding the influence of gut microbiota on neurodegenerative diseases and discussed the underlying mechanisms, relevant clinical implications, and potential applications.

## 2. Gut Microbiota and Neurodegenerative Diseases

Neurodegenerative diseases are characterized by the progressive loss of neuronal function, resulting in eventual motor and/or cognitive function impairment. Currently, the prevalence of neurodegenerative disease is rapidly rising. Although genetic susceptibility is a major risk factor of neurodegenerative diseases, environmental factors throughout one’s lifetime also exert a great influence on the onset, development and eventual severity of such diseases [30]. Currently, the increasing clinical and preclinical evidence suggests that changes in intestinal microorganisms may, to a certain extent, lead to an increased risk of neurodegeneration. Although the underlying mechanisms remain largely unknow, the hypothesis that gut microbes affect neurodegenerative diseases through the gut–brain axis is gaining increasing attention.

Alzheimer’s disease (AD) is one of the most prevalent neurodegenerative diseases and is characterized by diminishing neurons and synapses, as well as a progressively declining cognitive function. Both environmental factors and genetics are considered to contribute to its etiology [31]. However, recent studies suggest that the alteration of gut microbiota composition is also related to the onset and development of AD. In the works of Brandscheid et al. [32], it is suggested that the abundance of Firmicutes increased, whereas that of Bacteroidetes decreased in 9-month-old 5× FAD mice (experiment), as compared to 9-month-old WT mice (control). Yet, in a similar study, gut microbiota of similar-aged APP PS1 mice (about 8 months old) showed a decrease in both Firmicutes and Bacteroidetes when compared with WT mice [33], a result partially contradictive with the study of Brandscheid et al. Beyond animal experiments, Voet et al. [34] studied the composition of gut microbiota in AD patients by 16S ribosomal RNA gene sequencing, and detected a reduced total abundance and diversity of microbiota in patient guts, as compared to healthy controls. In another human trial, an increased abundance of *Escherichia* and *Shigella* in the intestinal tract were found in patients with cognitive impairments and amyloid depositions, aa decreased abundance of *Eubacterium rectale* and *Bacteroides fragilis* was also observed, together with significantly increased inflammatory markers in circulation [35]. Furthermore, according to a recent comparative study, human AD patients had a decreased microbial diversity, compared to both healthy controls and patients with merely mild cognitive impairments (MCI). Furthermore, the abundance of Gammaproteobacteria, Enterobacteriales and Enterobacteriaceae all increased from the control group to the MCI group, and from the MCI group to the AD group. Additionally, a significant correlation was discovered between the clinical severity score of AD patients and the number of gut bacterial species with altered compositions [36]. All of this evidence suggests a significant alteration of gut microbiota in both animal models and patients of AD.

Gut microbiota secrete over 100 different types of metabolites, yet, to date, most of them still have undefined roles in the pathogenesis of AD. Butyric acid, propionic acid and other short-chain fatty acids were found to affect the activation of microglia and astrocytes and helped to reduce the inflammation and aggregation of Aβ and tau in brain tissues [37,38]. Moreover, intestinal bacteria can also secrete a large number of lipopolysaccharide (LPS) and amyloid proteins. On one hand, these substances can directly enter the brain through the intestinal and blood–brain barriers. On the other hand, they can also induce a series of inflammatory reactions and increase the permeability of these barriers [39]. Additionally, the microbial amyloid protein produced by gut microbiota can also interact with the Toll-like receptor TLR2 to induce the activation of pro-inflammatory mediators such as interleukin (IL-17A, IL22), subsequently inducing an immune response and stimulating the production of the amyloid protein in the brain neurons [40].

Parkinson’s disease (PD) is another prevalent example of neurodegeneration. The results of one study on gut microbiota dysbiosis found significantly elevated levels of indican (a dysbiosis marker) in PD patients [41]. In a large-scale cohort study that included 72 PD patients and an equivalent number of healthy controls, the results of the high resolution 16S sequencing suggested that the amount of Prevotellaceae was reduced by 77.6% in PD patients. Notably, Prevotellaceae is a primary producer of mucin, which forms a barrier along the intestinal epithelium to defend against invading pathogens [42]. Moreover, as is suggested in a similar study, the abundance of bacteria that produce butyrate (a substance with an anti-inflammation effect), namely *Roseburia* and *Faecalibacterium* spp., significantly decreased in PD patients as compared to age-matched healthy controls. Intriguingly, the accumulating evidence linked PD with the accumulation of pathogenic α-Syn in the gastrointestinal tract, implicating a potentially novel etiology.

Aside from AD and PD, less common forms of neurodegenerative disease include Huntington’s disease, amyotrophic lateral sclerosis (ALS), and motor neuron disease, etc.

## 3. Functions of Gut Microbiota

### 3.1. Enhanceing Intestinal Epithelial Barrier

The epithelium barrier of the intestine is a primary defense mechanism that protects the body against environmental pathogens. The contents of the barrier include the epithelial junction adhesion complex and a layer of mucus with secretory IgA, as well as antimicrobial peptides [43]. Once the barrier is breached, bacteria and other pathogens can reach the submucous layer and induce inflammation [44,45,46]. Certain non-pathogenic food-borne bacteria species are beneficial to the function of the intestinal barrier, of which the exact mechanisms remain insufficiently understood, though the increased expression of tight junction signaling-related genes induced by certain bacteria types might be one of the proper explanations [47,48]. As another possible protective mechanism, the association between the altered level of pro-inflammatory cytokine and intestinal tract permeability was also intensively elaborated [49,50]. Certain types of *Lactobacillus* enhanced the expression of mucins in human intestinal cell lines, yet the protective effect depended upon the adhesion of *Lactobacillus* on single-layered cells and this might not be the case of in vivo situations [51,52]. Additionally, there is also evidence indicating that the extractives of *Lactobacillus* alone increased the expression of MUC2 in HT29 cells, irrespective of the adhering mechanism [53]. Therefore, certain indigenous microbiota may have enhanced the intestinal epithelium barrier by increasing the production of intestinal mucus.

### 3.2. Preventing Gastrointestinal Infection

Indigenous microbiota in the intestinal tract is the first line of defense against invading exogenous pathogens [54]. Specifically, the beneficial microorganism species contribute to the prevention of intestinal infection through mechanisms such as altering the pH value of the intestinal microenvironment, secreting anti-bacterial substances, and directly competing for adhesion sites or nutrition on the epithelium surface [55,56,57]. In clinical scenarios, antibiotics-related diarrhea occurs mostly when the treatment with antibiotics starts to substantially disrupt the natural balance between the intestinal microbiota subpopulations and leads to the proliferation of harmful bacterial types (e.g., *Clostridium difficile*). Additionally, compared to the placebo control group, the administration of probiotics reduced the incidence rate of antibiotic-related diarrhea by 60% [58]. Additionally, *Lactobacillus* GG (a probiotic species) significantly shortened the disease course of infectious diarrhea in infants and children [59]. Both results suggested that certain types of gut microbiota might have played a substantial role in the systematic reaction against gut-mediated infection.

### 3.3. Immunomodulatory Effects

Gut microbiota can influence the progression of certain diseases through a modulatory effect on the host’s immune responses [60]. Complex interactions can occur between gut microbiota and the surface of intestinal mucosa, which enhance the host’s cellular immune reaction and manifest as the activation of immune cells (i.e., macrophages and antigen-specific cytotoxic T-lymphocytes) and the release of assorted cytokines [61]. Specifically, *Lactobacillus salivarius* and *Bifidobacterium breve* are considered important bacteria species that contribute to the stabilization of the immune system [62]. Other probiotic gut bacteria species, such as *Lactobacillus plantarum*, *Bifidobacterium infantis*, and *Lactobacillus rhamnosus*, might be efficacious in preventing and mitigating allergy and auto-immune diseases (e.g., irritable bowel syndrome and inflammatory bowel disease, etc.) [63,64,65].

### 3.4. Nutritional Benefits

Gut microbiota enrich the nutrient sources of the host by synthesizing certain vitamins and assorted biologically active metabolites (e.g., short-chain fatty acids, SCFAs), most of which can be directly or indirectly utilized by the host system. Previous animal experiments reached this conclusion based on the fact that a number of vitamin types found in the intestines of regular mice could not be found in the intestines of germ-free mice [66,67]. Upon investigation, these particular vitamins are synthesized mostly by several specific bacterial types in the gut, including *Propionibacterium*, *Fusobacterium*, *Bacteroides*, and *Eubacterium* [67,68]. On the other hand, ingesting yogurt that contains *Lactobacillus bulgaricus*, *Lactobacillus delbrueckii*, or *Lactobacillus acidophilus* can reportedly reduce the incidence of lactose intolerance, presumably with the enzyme lactase within the bacteria per se [69]. Nevertheless, it is worth mentioning that the primary metabolic function of colonic microbiota is the fermentation of nondigestible carbohydrates, of which SCFA is one of the major end products. SCFAs have trophic functions that are fundamental in the life cycles of the intestinal epithelium [70].

## 4. Underlying Mechanisms of the Influence of Gut Microbiota on Neurodegeneration

The notion of maintaining gut microbiome homeostasis for the health of neurological system is attracting increasing attention. Here, we discuss the mechanisms through which gut microbiota exert an influence, both direct or indirect, on the central nervous system (Figure 2).

### 4.1. Production of Assorted Functional Metabolites

#### 4.1.1. Tryptophan Metabolites

The tryptophan (TRP)–kynurenine (KYN) pathway and its metabolites were observed to play an important role in neuroinflammation [71,72]. Tryptophan is one of the eight essential amino acids in the human body and can be attained from a dietary protein only; it is the only amino acid with an indole structure. The level of free tryptophan is determined by both food intake and the activity of three tryptophan metabolic pathways. Of the three, the indole pathways are directly regulated whereas the kynurenine pathway and serotonin pathway are indirectly regulated by gut microbiota. Overall, a rather small fraction of free tryptophan (TRP) is utilized for the synthesis of proteins and the production of neurotransmitters, e.g., serotonin 5-HT)/neuromodulators such as tryptamine, whereas over 95% of free tryptophan is the degradation substrate of the TRP-KYN pathway [73,74,75], where various bioactive metabolites such as neuroprotective antioxidants and neuroprotectants, toxic oxidants, and neurotoxins, as well as immunomodulators, are synthesized [71]. Notably, two key intermediate metabolites of the TRP-KYN pathway are quinolinic acid (QA) and kynurenine uric acid (KA). QA can induce neurodegeneration through NMDA-mediated excitotoxicity [76], while, as an endogenous NMDA receptor antagonist, KA can regulate the neurotoxic effect of QA and bears neuroprotective functions [77]. According to several studies, the TRP-KYN pathway is crucial for neurodegeneration as well as severe brain injury [78]. Being the rate-limiting enzymes of the pathway, hepatic tryptophan 2,3-dioxygenase (TDO) and extra-hepatic indoleamine 2,3-dioxygenase (IDO) also activate the TRP-KYN pathway, producing neuroactive metabolites such as QA and KA. Furthermore, an indicator of IDO/TDO activity is the kynurenine per tryptophan quotient (KYN/TRP) ratio. The evidence suggests that the increase in the KYN/TRP ratio is directly proportional to the severity of the impairment on cognitive function [79]. Additionally, there is also evidence that gut microbiota can directly influence the activities of key enzymes in the TRP-KYN pathway. It is worth mentioning that the activity of IDO is decreased in the intestinal tract of germ-free mice, yet it could be restored to normal by colonizing microorganisms in the intestinal tract immediately after weaning [80,81].

In recent years, the accumulating studies have shown that the serum concentration of kynurenine (KYN) and 3-hydroxykynurenine (3-HK), which are intermediate metabolites of TRP-KYN pathway, increased significantly in AD animal models; whereas, the concentrations of TRP and KA showed a downward trend, which was closely related to cognitive function impairment [82]. In Parkinson’s disease, the concentration of 3-HK in the frontal cortex, putamen and substantia nigra increased significantly, while the concentration of KA decreased [83]. On the other hand, the TRP-KYN pathway metabolites were also considered to be related to the pathological process of Huntington’s disease. Additionally, the concentrations of QA and 3-HK increased in both the neostriatum and cortex of Huntington patients and transgenic mice, compared to control group [84]. Interestingly though, the probiotic treatment alters the kynurenine levels [85]. Additionally, other studies have shown that as an inhibitory neurotransmitter, serotonin can reduce the formation of the Aβ plaque and regulate cognitive function [86]. It is worth noting that about 90% of serotonin is produced in the chromaffin cells of the gastrointestinal tract. Additionally, *Escherichia coli* and *Enterococcus*, which are common bacteria in the intestine, can also produce serotonin [87]. Therefore, gut microbiota may affect the function of the central nervous system by controlling the production of serotonin.

#### 4.1.2. Short-Chain Fatty Acids

SCFAs are one of the metabolic end products of gut microbiota. SCFAs are primarily composed of butyrate, propionate, and acetate, which are synthetized from undigested food carbohydrates and proteins [88]. The specific types of SCFAs produced by gut microbiota depend primarily on the relative amount of microbiota subgroups, i.e., microbiota composition. For example, microbes in the Firmicutes predominantly produce butyrate whereas *Bifidobacteria* spp. mostly synthetize lactate and acetate [89]. Regarding the functions of SCFAs, these small molecules participate in the cellular signaling of the epithelium via FFAR2 (G-protein-coupled free fatty acid receptors 2) and FFAR3 in the gastrointestinal tract. Meanwhile, SCFAs can also enter the systematic circulation passively or actively, exerting a rather broad range of physiological effects [90], including participation in the metabolism of glucose and lipids [91,92,93].

SCFA can affect the function and development of the nervous system. For example, SCFA was proved to aggravate the motor symptoms in sterile PD mice [94], yet they improved the recovery of experimental stroke mice [95]. Acetate is shown to be able to penetrate the blood–brain barrier and reduce eating behavior in mice [96,97]. Butyric acid is a multi-functional molecule, which plays a beneficial neuroprotective role and improves the health of the brain. In addition to being an important substrate for energy production, butyric acid can also increase the mitochondrial respiration rate and ATP production, inhibit histone deacetylase, and affect the function of many genes and a large number of cellular proteins [98]. In addition, short chain fatty acids can stimulate the release of neuropeptides by binding to homologous receptors such asGPR43, GPR41, peptide YY (PYY) and GLP-1 (Glucagon-like peptide 1), which play a special role in intestinal endocrine signaling. Once released, these peptides affect the regulation of energy homeostasis by activating intestinal and primary afferent vagal pathways [99].

#### 4.1.3. Histamine

In the intestinal tract, histamine is produced primarily by enterochromaffin (EC) cells. It plays significant roles in assorted physiological activities such as cell proliferation, wound healing, allergic reaction, immune cell regulation, etc., and functions as one of the most important neurotransmitters in the brain [100]. Large amounts of histamine receptors are found on the neurons in the hippocampus, thalamus, striatum, substantia nigra, and other brain regions, implicating their wide-ranging effects in the entire central nervous system. Recently, histamine was proposed to be a potential drug for the treatment of neurodegenerative diseases, specifically for MS and AD [101].

Histamine was recently found to be the metabolic product of intestinal microbiota. *Lactococcus*, *Lactobacillus*, *Streptococcus*, and *Pediococcus* all carry the gene of histidine decarboxylase, and thus can produce histamine [102]. *Lactobacillus reuteri*, previously considered as immune, modulatory, probiotic bacteria, were found to be able to transform the food-borne L-histidine into histamine [103]. As with neurodegenerative disorders, the increase in histamine levels is found to be associated with AD, and is believed to increase the level of nitrogen oxide, which is a stimulating factor of neural inflammation [104]. Furthermore, there are also studies that report the histaminergic signaling deficiency in vascular dementia rats [105]. In other words, histamine has a broad-ranging effect on the development of neurodegenerative disorders. Therefore, the regulation of its metabolism by gut microbiota can be a novel and potentially effective therapeutic approach for neurodegeneration.

#### 4.1.4. Ghrelin

Ghrelin is a neuropeptide that is generated in the gastrointestinal tract and transmits satiety signals; it can also be found in the central nervous system. Ghrelin is secreted primarily when the stomach is empty, after which it reaches the brain through blood circulation and generates the feeling of hunger. Moreover, ghrelin acts as a key modulating factor in multiple metabolism processes, including energy homeostasis, inflammation, and neuro-modulation [106,107]. Notably, ghrelin is proven to be neuro-protective in both AD and PD [108]. The evidence suggests that the expression of ghrelin is reduced in the brains of AD patients, indicating a critical role in the pathological process of AD [109]. In PD, the activation of the ghrelin receptors of substantia nigra neurons stimulates the expression of tyrosine hydroxylase, promoting the synthesis of dopamine. Furthermore, it was demonstrated that, upon administering *Bifidobacterium* spp., the secretion of ghrelin in the human system was significantly decreased [110]. In brief, ghrelin produced from gut microbiota has a significant role in maintaining the regular function of the brain and is a promising target for the treatment of neurodegeneration.

#### 4.1.5. Neuro-Metabolites

A large number of neuro-metabolites are secreted directly by gut microbiota, or by secretory epithelial cells due to the stimulation of microbiota. These neuro-metabolites include neurotransmitters that act directly on the central nervous system (CNS), signaling cascades or other signaling pathways and exerting direct or indirect influences on the regular function of the CNS [111,112]. For example, *Lactobacillus* and *Bifidobacterium* strains can produce large amounts of GABA when a suitable substrate is present [113]. Gut-derived neuro-metabolites communicate with the central nervous system by stimulating the local afferent vagal fibers or through distal endocrine action. The variations in the level of neurotransmitters lead to changes in behavior, such as the heightened activity of spontaneous motor nerves due to elevated levels of noradrenaline, dopamine, and serotonin in the striatum [113]. Additionally, this phenomenon is significant in the management of neurodegenerative diseases where there is, more often than not, a dysregulation of neurotransmitter production that ultimately contributes to disease progression.

### 4.2. Microbial-Associated Molecular Patterns (MAMPs) and Immune Responses

Microbial-associated molecular patterns (MAMPs) are highly conservative components of assorted microbials [114]. They are the essential components that compose microbial pathogenicity [115]. MAMPs bind to the pattern recognition receptor (PRR) in the immune cells and trigger the secretion of inflammatory cytokines. These cytokines can affect the brain indirectly through the peripheral vagus nerve, or directly through the permeable areas of the blood–brain barrier [116]. Peptidoglycan and LPS are two primary constituents of MAMP [117]. Peptidoglycans are structural components of almost all bacterial cell walls and can be transferred to the developing brain, exerting an influence on gene expression and social behavior [118]. LPS, another ubiquitous surface molecule of Gram-negative bacteria, has been shown to induce cognitive impairment in mice upon injection [119], and also reportedly affected the development of the fetal brain [120,121]. The existence, structure, and immunomodulatory activity of MAMPs vary with different bacterial species. Therefore, gut microbiota may affect the host’s exposure level and response to specific MAMPs by restoring normal bacterial flora constitution, and thus inhibit neuroinflammation and regulate the host’s overall health status and behavior changes.

### 4.3. Vagus Nerve

The vagus nerve, or the tenth cranial nerve, transmits sensory information between the peripheral and central nervous systems, and functions as a direct connection between the gut and the brain [122,123]. Multiple studies have shown that the primary afferent fibers of the vagus nerve mediate the communication between gut microbiota and the central nervous system [124,125]. The pathogens and certain indigenous microbiota in the gut affect brain function and behavior by activating vagus nerve neurons, and thus altering neural activity [122,126,127]. These responses all disappeared after the severance of the vagus nerve. However, the specific bacterial metabolites that mediate these effects remain undefined. Exploring the role of the vagal afferent pathway in regulating the crosstalk between gut microbiota and the brain may pave the way for microbiota-related therapy to be tested in the treatment of neurological diseases.

### 4.4. Inhibition on Harmful Gut Microorganisms

Gut microbiota can protect host systems from possible infections by directly inhibiting intestinal pathogens. Generally, such inhibition effects are realized primarily through the production of anti-bacterial substances such as hydrogen peroxide, bacteriocins, and organic acids. Furthermore, certain native bacterial species exert modulatory effects on the pathogenicity of intestinal pathogens, mechanisms of which include anti-colonization n and toxin neutralization, etc. Furthermore, gut microbiota’s competition for nutritional substances also exerts a suppressive effect on harmful gut microorganisms [128,129].

### 4.5. Others

Bacteria is occasionally capable of passing through the blood–brain barrier, or the blood–cerebrospinal fluid barrier, to enter the central nervous system, the mechanisms of which include trans-cellular infiltration, paracellular entering, or via the infected leukocytes [130]. Branton et al. [131] detected the existence of bacteria in the brain tissue of multiple sclerosis (MS) patients and discovered that Proteobacteria were the dominant flora in the cerebral white matter of female MS patients, a phenomenon that is reportedly associated with the expression of inflammation-related genes in patients’ brains. Furthermore, the research confirmed a strong interaction between the bacterial presence in the brain and host responses involving NF κB-related signaling, a pivotal pathway in neuroinflammation and MS pathogenesis, in demyelinating lesions [132]. Studies also found that *Porphyromonas gingivalis*, a primary pathogen of chronic periodontitis, existed in the brain tissue of patients with Alzheimer’s disease. Additionally, Porphyromonas gingivalis was also associated with impaired spatial/episodic memory in AD [133]. In related animal experiments, these pathogenic bacteria “invade” the brain after an oral infection, leading to an increase in the amyloid beta protein [134]. However, it is still not clear whether the gut microbes can enter and colonize the brain simply through blood circulation, as the evidence for this is very limited. In a study presented at the 2018 annual meeting of the American Society of Neuroscience, 34 brain samples of healthy people and schizophrenic patients were analyzed by an electron microscope, and bacteria were found in all samples. In the same study, bacteria were also found in the brains of healthy mice but not in sterile mice. These bacteria mainly came from three common intestinal bacteria phylum (Firmicutes, Proteobacteria, and Bacteroidetes), which tended to gather in the astrocytes and neuron axons adjacent to the blood–brain barrier and did not seem to have caused the inflammation of the mouse brain tissues [135]. After a thorough inspection of these results, the possibility of experimental contamination is yet to be excluded, and further studies are still needed to support these preliminary findings.

## 5. Microbiota-Associated Therapy in Application

Two primary forms of microbiota-associated therapy in the current applications of neurological diseases are probiotics therapy and fecal microbiota transplantation [136].

### 5.1. Probiotics Therapy

As demonstrated by a number of animal experiments, the administration of beneficial microorganisms (i.e., probiotics) could be a promising method to prevent and treat neurodegenerative diseases. Particularly, the probiotic preparations for oral administration, whether for single-strain or multi-strain microorganisms, proved to be a successful therapeutic strategy. For example, Probiotic-4 is a preparation containing *Bifidobacterium lactis*, *Lactobacillus casei*, *Bifidobacterium bifidum*, and *Lactobacillus acidophilus*. It functions to significantly inhibit Proteobacteria (phylum), *Pseudomonas* (genus) and the *Lachnospiraceae*_NK4A136_group (genus); meanwhile, it significantly reduces the ratio of Firmicutes to Bacteroidetes and improves the cognitive function of aged SAMP8 mice through the inhibition of the NF-κB signaling pathway and the inflammatory reactions, which are mediated by TLR4 and RIG-I [137]. In another study, a probiotic mixture containing *Lactobacillus acidophilus*, *Lactobacillus fermentum*, *Bifidobacterium lactis*, and *Bifidobacterium longum* was given to rats injected with Aβ for 8 weeks. The results indicated an improved spatial memory, learning impairments and a reduced oxidative stress through the alteration of the gut microbiota composition [138]. Recently, SLAB51, a formulation composed of *Lactobacillus* and *Bifidobacterium*, was reported to modulate the gut microbiota of 3xTG AD mice, increasing the relative abundance of *Bifidobacterium* spp. while reducing the relative abundance of *Campylobacter*. The study concluded that these alterations of microbiota composition, together with the abundance of SCFA in the intestine and the increased level of the neuroprotective, intestinal peptide hormone in plasma, could alleviate the decline of cognitive ability by reducing the Aβ aggregate and, subsequently, brain damage [139]. The combination of *Lactobacillus helveticus* R0052 and *Bifidobacterium longum* R0175 could significantly decrease the levels of pro-inflammatory cytokines in the serum and hippocampus that were induced by LPS, reducing the apoptosis of hippocampal cells, and attenuating the adverse effects of LPS on memory by expressing BDNF proteins [140,141]. In another study, ddY mice injected with Aβ were treated with 1 × 10^9^ CFU *Bifidobacterium breve* A1, which improved both the behavior and memory of the mice, whilst inhibiting the expression of the immune response genes and inflammation-related genes in the hippocampus [142]. Another study suggested that the treatment of *Lactobacillus plantarum* MTCC1325 for 60 days in a D-galactose-induced rat model not only improved cognitive impairment, but also restored the level of ACh and the features of histopathology to the control levels [143]. Wang et al. [144] and Liang et al. [145] found that *Lactobacillus fermentum* NS9 and *Lactobacillus helveticus* NS8 alleviated ampicillin-induced spatial memory impairment and improved the spatial memory of chronic restraint stress. Additionally, a recent study showed that exercise combined with a probiotics mixture could reduce Aβ plaque in the hippocampus, improving cognitive ability and, ultimately, slowing down the development of AD in an APP/PS1 mouse model [146]. Zhang et al. reported that, after a treatment with 2% butyrate (a natural bacterial product that helps restore the homeostasis of gut microbiota), the gut microbiota balance and intestinal epithelial barrier integrity of G93A mice was reinstated. Additionally, aside from improving the central and peripheral symptoms of the disease, G93A mice demonstrated a prolonged survival time and reduced body weight [147].

Similar to probiotics applications in humans, *Lactobacillus* and *Bifidobacterium* are widely used in probiotic preparations in clinical settings, as bacteria members of these two categories are widely used to promote human health and are rated as GRAS (generally regulated as safe) for human consumption [148]. In the study by Akbari et al., patients with AD were fed with *Lactobacillus acidophilus*, *Lactobacillus* casei, *Bifidobacterium bifidum*, and *Lactobacillus* fermentum for 12 weeks (200 mL/D). Compared with the control group, the probiotics treatment had no significant effect on oxidative stress, inflammation, fasting, blood glucose, and biomarkers of lipid distribution, but had a positive effect on the cognitive function and insulin metabolism of AD patients [149]. However, as reported by Agahi A et al. [150], after AD patients were given *Lactobacillus fermentum*, *Lactobacillus plantarum*, and *Bifidobacterium lactis*, or *Lactobacillus acidophilus*, *Bifidobacterium bifidum*, and *Bifidobacterium longum* (3 × 10^9^ CFU), respectively, for 12 weeks, no difference was observed regarding the cognitive test scores between the two groups, and the serum inflammatory factors (IL-6, IL-10 and TNF-α), glutathione (GSH), malondialdehyde (MDA), and itric oxide (NO) were not significantly changed either, suggesting that probiotics could not effectively improve the cognitive and biochemical indicators of patients with severe AD. Therefore, in addition to the formulation and dosage of probiotics, the severity of the disease per se also plays a critical role in causing the beneficial effect of probiotic intervention in AD patients. According to Kobayashi Y et al., after taking a *Bifidobacterium breve* A1 capsule for 12 weeks, the scores of a Mini Mental State Examination (MMSE) and a Repeatable Battery for the Assessment of Neuropsychological Status (RBANS) both changed significantly, while the changes in the levels of serum lipid, inflammation, and oxidative stress markers were not significant, suggesting that *Bifidobacterium breve* A1 was safe and could improve the impaired cognitive function (i.e., memory impairment) of the elderly [151]. Furthermore, the consumption (for 4 weeks) of a fermented milk containing multiple probiotic strains and probiotic fibers was shown to improve PD complications, particularly the symptom of constipation [152].

The safe use of probiotics was strongly supported by various studies demonstrating how endogenous microorganisms played important roles in our personal health, particularly regarding neurodegeneration (Table 1). These studies showed that positive results could be obtained when specific microbial strains were administered as probiotics.

### 5.2. Fecal Microbiota Transplantation (FMT)

FMT refers to the transplantation of feces containing gut microbiota from healthy donors to recipients with dysbacteriosis, by means of an enema or nasogastric, nasointestinal, or endoscopic approaches, aiming to restore the normal diversity and functionality of the gut microbiome [157,158]. This method is currently considered an effective treatment for the recurrent infection of *Clostridium difficile* [159]. With the bidirectional signal interaction of the gut–brain axis, FMT is also considered as a potential treatment for certain extraintestinal diseases, including neurodegenerative diseases. Dodiya et al. [160] found that the transplantation of fecal microbiota from APP/PS1-21 male mice into age-matched antibiotic-treated APP/PS1-21 male mice could restore the normal intestinal microbiome, and partially reverse Aβ pathology and microglia morphology. Another study found that, compared with healthy control mice, mice transplanted with feces from patients with AD had a poorer cognitive function and fewer fecal metabolites, which were associated with the nervous system (e.g., GABA, taurine, and valine) [161]. Admittedly, there are currently few studies reporting the effect of FMT on neurological diseases in human subjects, and safety is still a major issue in translating FMT research into human trials. As with existing or ongoing human studies, fecal donors and samples of FMT are subjected to examinations for potentially pathogenic bacteria, viruses, and parasites, etc. [162]. Nevertheless, the exact and optimal microbial composition of the samples to be transplanted is still under investigation, which, as far as human trials are concerned, not only raises potential safety concerns, but also leads to problems with the interpretation of results.

### 5.3. Future Prospect

Neurodegenerative diseases have complex conditions that usually involve cognitive, motor, and systemic dysfunctions. Both genetic and environmental factors are considered relevant to their pathogeneses, among which gut microbiota may also be a potential influencing factor. The manipulation of intestinal microbiota by FMT may affect the symptoms or progression of diseases through immune, endocrine, metabolic, and/or neural pathways mediated by intestinal microorganisms, thus constituting, despite limited evidence, a potential treatment option for a variety of neurological diseases. Future studies should carefully weigh up the potential benefits and risks of microbiota-related interventions and put safety first regarding human subjects. Compared with FMT, probiotic therapy is more selective and targeted. Probiotics can directly or indirectly act against or destroy other harmful microorganisms in the intestine. At the same time, it is necessary to ensure that these probiotics maintain a normal activity and metabolism in the host body. Probiotics are often exposed to gastric juice and bile before they reach the intestinal tract. These acidic environments lead to the death of most bacteria before they even reach the small intestine or colon. Therefore, effective measures should be taken to ensure the effective activity of probiotics. In summary, a better understanding of the gut–brain axis is expected to accelerate the development of probiotic therapy, as well as the prevention of and intervention in neurodegenerative diseases.

## 6. Conclusions

Gut microbiota are implicated in the pathogenesis of neurodegenerative diseases. Metabolites produced by intestinal microorganisms, as chemical messengers, mediate the interaction between microbiota and the host. Of note, some of these metabolites are proven to affect the outcome of neurodegeneration. However, there is still much to explore regarding the functions of the metabolites of gut microbiota. Our ultimate goal with this review was to summarize the existing evidence connecting gut microbiota with neurodegenerative diseases, which may potentially facilitate researchers in the field of neurodegeneration who are exploring new possibilities in the largely uncharted territory of gut microbiota. However, one particular limitation of this review is its narrow scope with an intensive focus on AD and PD. The application of different animal models and an omics analysis may help us to further discover and decipher the relationship between intestinal microorganisms, metabolites, and neurodegenerative diseases. The current pre-clinical studies and human clinical trials of intestinal microbiota are still in the early stages, but many studies have pointed out the potentially significant role of various microbiota-related treatments (e.g., probiotics and FMT) in changing the composition of gut microbiota. Further studies on the relationship between intestinal microorganisms and certain important neuro-metabolites will hopefully provide new concepts and methodologies for the prevention of and intervention in neurodegenerative diseases.

## Figures and Tables

**Figure 1 microorganisms-09-02281-f001:**
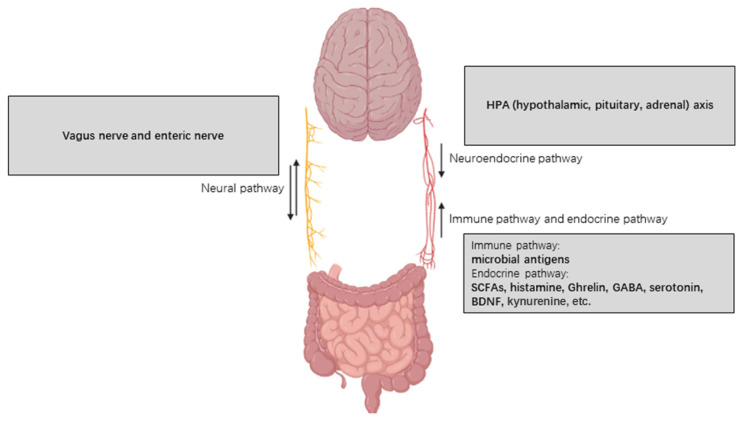
Pathways and participants of gut–brain axis. The gut–brain axis comprises primarily of a neural pathway, neuroendocrine pathway, immune pathway and endocrine pathway, of which the neural pathway functions through vagus and enteric nerves, whereas the latter three pathways function through blood circulation.

**Figure 2 microorganisms-09-02281-f002:**
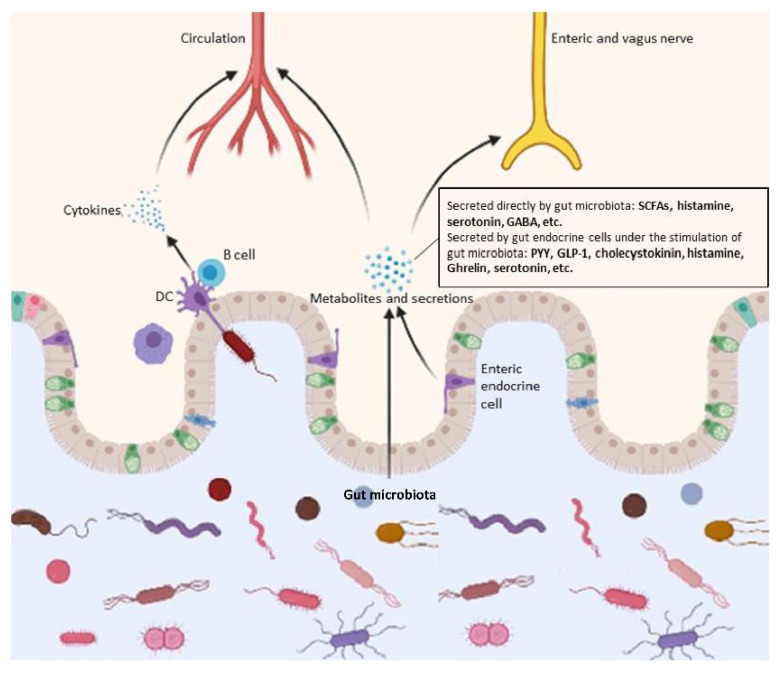
Mechanisms through which gut microbiota exert an influence on neurodegenerative diseases. Shown in the figure are the structural bases (i.e., blood circulation, and enteric and vagus nerves) and major events during the process.

**Table 1 microorganisms-09-02281-t001:** Probiotics administration influences gut microbiota and the gut–brain axis: evidence from clinical and pre-clinical studies.

Probiotic Used	Subjects/Samples	Function	Diseases Involved	Reference
*Lactobacillus plantarum* MTCC1325	D-galactose-induced AD albino rats (3 months old)	Reduced formation of Aβ plaques, restored Acetylcholine leve, improved cognitive function	AD	[143]
*Bifidobacterium breve* strain A1	Aβ-injected male ddY mice (10 weeks old)	Produced SCAFs, regulated immune responses and inhibited neural inflammation, improved cognitive function	AD	[142]
*Lactobacillus fermentum* NS9	Ampicillin induced male SD rats	Restored normal composition of gut microbiota, and reversed antibiotics-induced anxiety behavior and spatial memory defects	AD	[144]
*Lactobacillus helveticus* NS8	Adult male SD rats to construct depression-like rat model of chronic restraint stress	Restored level of 5-HT and BDNF in hippocampus body, regulated inflammation responses	AD, anxiety, depression	[145]
	Male SD rats to construct rat model of hyperammonemia	Reduced the level of inflammation biomarkers, decreased 5-HT metabolism, restored cognitive function, improved anxiety-like behavior		[153]
*Bifidobacterium breve* strain A1	Elderly with memory dysfunction	Improvement with cognitive function	AD	[150]
*Bifidobacterium infantis* 35624	SD rats	Modulated HPA stress response, reduced pro-inflammation immune response, increased level of 5-HTP	Depression	[85]
*Lactobacillus reuteri* ATCC 23272	C57BL/6 and BALB/c mice	Inhibited metabolism of tryptophan/kynurenine	Depression	[154]
*Lactobacillus reuteri* ATCC-PTA-6475	ASD mice	Upregulated level of oxytocin in brain, regulated plasticity of neurons	ASD	[122]
*Bifidobacterium longum* NCC3001	AKR mice	Upregulated level of brain-derived neurotrophic factor, regulated plasticity of neurons	Anxiety	[126]
*Lactobacillus helveticus* R0052 and *Bifidobacterium longum* R0175	LPS-induced rats	Reduced level of pro-inflammatory cytokines, reduced the apoptosis of hippocampal cells, improved memory	AD	[140,141]
*Lactobacillus rhamnosus* GG(L-GG), *Bifidobacterium animalis lactis*(BB-12), and *Lactobacillus acidophilus (LA-5)*	MPTP-induced mice	Butyrate, prevented the loss of dopaminergic neurons by upregulating neurotrophic factors and inhibiting the expression of Mao B	PD	[155]
DW2009: a mixture of fermented soybean powder and *L. plantarum* C29 freeze-dried powder.	MCI patients	Increased the abundance of Lactobacilli, increased serum BDNF level, improved cognitive function	AD	[156]

AD Alzheimer’s disease, PD Parkinson’s disease, ASD Autism Spectrum Disorder, SCAFs short-chain fatty acids, 5-HT 5-hydroxytryptamine, 5-HTP 5-hydroxytryptophan, BDNF brain-derived neurotrophic factor, HPA The hypothalamic–pituitary–adrenal axis, LPS lipopolysaccharides, MPTP Pyridine,1,2,3,6-tetrahydro-1-methyl-4-phenyl-, Mao B monoamine oxidase B, MCI mild cognitive impairment.

## Data Availability

Not applicable.

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
