# Peer review of "Gut Microbiota Regulation and Their Implication in the Development of Neurodegenerative Disease"

_microorganisms, 2021, doi:10.3390/microorganisms9112281_

Round 1

Reviewer 1 Report

New insights are shifting toward an important role of microbiota in several diseases. This study reviews preclinical and clinical studies highlight the relation of gut microbiota and neurodegenerative diseases and also refers to treatment under study based on microorganisms. I suggest a minor revision of the manuscript. The main arguments are listed below:

  1. There are many mistakes in the spelling of the names of the microorganisms: line 111: Faecalibacterium, line 126 y 130: Lactobacillus, line 223: E.coli y Enterococcus, line 444: C. difficile
  2. Figure 2 is not very descriptive of the mechanism.
  3. In lines 252 and 290 EC cells and CNS are not defined. The authors use the abbreviations.
  4. Huntington’s disease was not cited in the introduction despite review about description gut microbiota it is done later.
  5. certain references are misquoted as they include the initial of the name (line 401 and 409).

  6. all the abbreviations included in table 1 should cited in the legend. 
  7. it would be more interesting to reorganise section 5.1 according to the neurodegenerative disease

Author Response

Response to Reviewer 1

Q1. There are many mistakes in the spelling of the names of the microorganisms: line 111: Faecalibacterium, line 126 y 130: Lactobacillus, line 223: E.coli y Enterococcus, line 444: C. difficile.

R: Sincere gratitude to you for your meticulous advice. We have carefully checked and corrected the spelling of the names of the microorganisms.

Q2. Figure 2 is not very descriptive of the mechanism.

R: Thank you for your kind suggestion. Please allow us to introduce our emphasis. The primary purpose of fig 2 was to describe the two mechanisms (i.e. via blood, and via nerve) through which gut microbiota affects brain. To highlight these two pathways, we intentionally omitted some less relevant details. However, we also noticed that adding the words “gut microbiota” at the start of the lowest arrow in the figure can make the whole picture seem more clarified. Thus we provided a revised version of this figure, with the added words.

Q3. In lines 252 and 290 EC cells and CNS are not defined. The authors use the abbreviations.

R: Thanks a lot for the reminder. And we have added the full names of the abbreviations.

Q4. Huntington’s disease was not cited in the introduction despite review about description gut microbiota it is done later.

R: Sorry for the careless mistake. According to your advice, we have added a sentence mentioning other less common forms of neurodegenerative diseases, as a prelude for the subsequent contents regarding Huntington’s disease.

Q5. certain references are misquoted as they include the initial of the name (line 401 and 409).

R: We were grateful for your kind advice. Accordingly, we have meticulously gone through the whole manuscript de novo to have made sure that no further references are misquoted now.

Q6. all the abbreviations included in table 1 should cited in the legend.

R: Sorry for the inconvenience. All abbreviations in Table 1 are now fully denoted in the legend below the table.

Q7. it would be more interesting to reorganise section 5.1 according to the neurodegenerative disease.

R: Thank you so much for this pointed advice. We understand that section 5.1 might seem a little unorganized and uneasy for readers to digest. To clarify our purpose, we aimed to elaborate these contents in the order of: (1) utilization of probiotics in animal models, (2) utilization of probiotics in humans. As most researches referenced in this section were all regarding AD, therefore we had not endeavored to organize the contents according to specific disease types. And we sincerely hope for your understanding.

Reviewer 2 Report

28 September 2021

Review on the manuscript titled “The Gut Microbiota Regulation and its Implication in the Development of Neurodegenerative Disease” by Sun P et al., submitted to Microorganisms

Manuscript ID: microorganisms-1389576

Dear Authors,

The gastrointestinal microbiota plays an important role in the brain function through the gut-brain axis. The authors review the role of gastrointestinal microbiota in neurodegenerative diseases, discussing the application of microbiota for the treatment of neurodegenerative diseases.

Please reconsider the following parts:

  1. Page 1, Abstract: Please clearly describe the background of the topic and the purpose of the manuscript.
  2. A graphic abstract summarizing the manuscript is highly recommended.
  3. Page 2, Keywords: Please list more keywords up to ten.
  4. Page 1, Introduction, Paragraph 1: Psychiatric disorders are missing.
  5. Page 1, Introduction, Paragraph 2: “… which primarily includes neuroendocrine …” Metabolic is missing.
  6. Page 1, Lines 36-39: It deserves to include kynurenines.
  7. Pages 1,2, Introduction: Please write the manuscript discusses Alzheimer’s disease and Parkinson’s disease.
  8. Page 1 Figure 1: Please include metabolites and add short description in the caption.
  9. Page 3, The Section 2, Paragraph 1: The environmental factors may include cyanobacterium. Please cite references. Suggested reference: https://doi.org/10.3390/ijms22168726.
  10. Pages 3, The Section 3.1: The involvement of inflammation in neurodegenerative diseases is reviewed recently. Suggested references: https://doi.org/10.3390/ijms21072431.
  1. Pages 3,4, The Section 3: Please expand the section, especially 3.3. and include the roles of the microbiota in the production of certain metabolites.
  2. Page 5, Figure 2: Please add short description in the caption.
  3. Page 5, The Section 4.1.1: Please add short general description of the disturbance of tryptophan-kynurenine metabolism in various diseases in the beginning of the section. Suggested references: doi: 10.1016/j.neubiorev.2020.08.010; 10.1038/s41598-020-73918-z; doi: 10.17219/acem/139572; https://doi.org/10.3390/biomedicines9040340.
  4. Page 5, Lines 192-193: There are several bioactive besides kynurenic acid and quinolinic acid. Please describe briefly. Suggested reference: https://doi.org/10.3390/biomedicines9080897.
  5. Page 5, Lines 196-200: Some kynurenine metabolites do not pass the blood-brain barrier. Please describe it briefly. Suggested reference: https://doi.org/10.3390/ijms21249338; https://doi.org/10.3390/ijms21176045.
  6. Pages 5, Lines 204-205: The relationships between kynurenines and symptoms including cognition are discussed recently. Suggested references: https://doi.org/10.3390/biomedicines9070734.
  7. Pages 7,8, The Section 4.2: The interaction of misfolded protein and the microbiota in neuroinflammation was discussed recently. Suggested reference: https://doi.org/10.3390/cells9112476.
  8. Page 8, Lines 348-351: The abundance of Porphyromonas gingivalis and the level of serum antibody in Alzheimer’s diseases is reported rececntly. Suggested reference: https://doi.org/10.3390/biom11060845.
  9. Page 9, The Section 5: Please present short general description regarding the microbiota-related intervention. Suggested reference: https://doi.org/10.3390/brainsci11081038.
  10. Pages 11,12, The Section 5.3: Please describe weaknesses or limitation in the present review, potentials, the ultimate goal, research or knowledge needed to achieve, the biggest challenge in this goal, and future research direction, among others.

The manuscript contains two figures, one table, and 154 references. The manuscript carries important value presenting the role of gastrointestinal microbiota in neurodegenerative diseases and possible interventional targets. However, the manuscript deserves to be revised as suggested above to improve the quality of presentation. I reconsider this manuscript for publication after major revision.

Best regards,

Author Response

Response to Reviewer 2

Q1. Page 1, Abstract: Please clearly describe the background of the topic and the purpose of the manuscript.

R: Our sincere gratitude to you for your helpful suggestion. According to your advice, we have re-arranged the abstract, with the background described more clearly and the purpose more specified.

Q2. A graphic abstract summarizing the manuscript is highly recommended.

R: We really appreciate your constructive suggestion. Yet due to the fact that this review comprises mainly of 4 separate and relatively independent parts, which are: (1) association of gut microbiota with neurodegenerative disease, (2) functions of gut microbiota, (3) mechanisms of gut microbiota’s function, (4) application of microbiota-related therapy, it is thus very difficult and impractical to summarize and integrate all these contents in one single graphic abstract. We therefore sincerely hope for your understanding on this particular issue.

Q3. Page 2, Keywords: Please list more keywords up to ten.

R: According to your advice, key words were added with total number of key words increased to 10.

Q4. Page 1, Introduction, Paragraph 1: Psychiatric disorders are missing.

R: Thank you so much for the reminder. We have added “psychiatric disorders” with proper citation.

Q5. Page 1, Introduction, Paragraph 2: “… which primarily includes neuroendocrine …” Metabolic is missing.

R: It’s very kind of you to have reminded us of this issue. But we chose to omit “metabolic” because, in this setting, as metabolites functions by entering blood circulation, we therefore believe “endocrine pathways” is sufficient to have included “metabolic pathway”. We sincerely wish for your understanding on this particular issue.

Q6. Page 1, Lines 36-39: It deserves to include kynurenines.

R: Sorry for the careless mistake. We have now added “kynurenines” with relevant citation.

Q7. Pages 1,2, Introduction: Please write the manuscript discusses Alzheimer’s disease and Parkinson’s disease.

R: We really appreciate your kind and pointed suggestion. Please allow us to explain our intentions. In the first place, we discussed Alzheimer’s disease and Parkinson’s disease in the Introduction section. But later we felt that it is most appropriate to split it from the Introduction section and arrange a specialized section (i.e. section “2. Gut Microbiota and Neurodegenerative Diseases”) for these discussions. We look forward to your understanding on this issue.

Q8. Page 1 Figure 1: Please include metabolites and add short description in the caption.

R: This advice is very helpful and we’re sorry for our neglect. As microbial metabolites reach and influence the brain through bloodstream, we therefore included all important microbial metabolites in the “endocrine pathway”, as is demonstrated in the figure. And we also revised the figure by adding kynurenine into the metabolites. We hope you could understand our perspective. Additionally, description was added in the figure caption.

Q9. Page 3, The Section 2, Paragraph 1: The environmental factors may include cyanobacterium. Please cite references. Suggested reference: https://doi.org/10.3390/ijms22168726.

R: Thank you for this valuable and convenient advice. Adding this reference is very helpful.

Q10. Pages 3, The Section 3.1: The involvement of inflammation in neurodegenerative diseases is reviewed recently. Suggested references: https://doi.org/10.3390/ijms21072431.

R: Thank you for this valuable and convenient advice. Adding this reference is very helpful.

Q11. Pages 3,4, The Section 3: Please expand the section, especially 3.3. and include the roles of the microbiota in the production of certain metabolites.

R: Sorry for your confusion regarding section 3.3. Please allow us to explain. In section 3.3, we primarily describe the interaction between gut microbiota per se and immune components in the host’s intestinal mucosa. Metabolites of microbiota has little role to play in this immune provoking process. Therefore, such contents describing the roles of the microbiota in the production of certain metabolites were not added in the first place. We sincerely wish for your understanding on this particular issue.

Q12. Page 5, Figure 2: Please add short description in the caption.

R: Sorry for the neglect. Description is now added in the caption.

Q13. Page 5, The Section 4.1.1: Please add short general description of the disturbance of tryptophan-kynurenine metabolism in various diseases in the beginning of the section. Suggested references: doi: 10.1016/j.neubiorev.2020.08.010; 10.1038/s41598-020-73918-z; doi: 10.17219/acem/139572; https://doi.org/10.3390/biomedicines9040340.

R: Thank you so much and this advice is very helpful indeed. We have added a short description with proper citation at Line 195, 196. However, we used another article from the journal Biomedicines (MDPI publishing) as the reference, which we thought would be more suitable considering its closely relevant content.

Q14. Page 5, Lines 192-193: There are several bioactive besides kynurenic acid and quinolinic acid. Please describe briefly. Suggested reference: https://doi.org/10.3390/biomedicines9080897.

R: This advice is very helpful and we really appreciated it. Accordingly, we have added a short description citing this reference.

Q15. Page 5, Lines 196-200: Some kynurenine metabolites do not pass the blood-brain barrier. Please describe it briefly. Suggested reference: https://doi.org/10.3390/ijms21249338; https://doi.org/10.3390/ijms21176045.

R: Thank you for this pointed advice. However, to avoid confusion in the first place, we decided to delete the sentence “Notably, kynurenine produced in the intestinal tract can effectively pass through the blood-brain barrier and directly promote the production of the above-mentioned metabolites in brain tissue.”

Q16. Pages 5, Lines 204-205: The relationships between kynurenines and symptoms including cognition are discussed recently. Suggested references: https://doi.org/10.3390/biomedicines9070734.

R: Thank you and we really appreciate this helpful advice. We have added this citation.

Q17. Pages 7,8, The Section 4.2: The interaction of misfolded protein and the microbiota in neuroinflammation was discussed recently. Suggested reference: https://doi.org/10.3390/cells9112476.

R: Thank you so much and we have added this citation.

Q18. Page 8, Lines 348-351: The abundance of Porphyromonas gingivalis and the level of serum antibody in Alzheimer’s diseases is reported rececntly. Suggested reference: https://doi.org/10.3390/biom11060845.

R: Thank you so much and this advice is very helpful. We have added a short description with this citation referenced.

Q19. Page 9, The Section 5: Please present short general description regarding the microbiota-related intervention. Suggested reference: https://doi.org/10.3390/brainsci11081038.

R: Thank you for this helpful advice. We have added a short general description and cited this useful reference as you have suggested.

Q20. Pages 11,12, The Section 5.3: Please describe weaknesses or limitation in the present review, potentials, the ultimate goal, research or knowledge needed to achieve, the biggest challenge in this goal, and future research direction, among others.

R: We really appreciate this helpful advice. Accordingly, we have added “limitation”, “potentials”, “ultimate goal”, and “future research direction” in the conclusion section.

Round 2

Reviewer 2 Report

19 October 2021

Review on the manuscript titled “The Gut Microbiota Regulation and its Implication in the Development of Neurodegenerative Disease” by Sun P et al., submitted to Microorganisms

Manuscript ID: microorganisms-1389576

Dear Authors,

The gastrointestinal microbiota plays an important role in the brain function through the gut-brain axis. The authors review the role of gastrointestinal microbiota in neurodegenerative diseases, discussing the application of microbiota for the treatment of neurodegenerative diseases.

The manuscript contains two figures, one table, and 163 references. The authors addressed their response properly and revised the manuscript accordingly. Thus, the quality of the manuscript is substantially improved. The manuscript carries important value presenting the role of gastrointestinal microbiota in neurodegenerative diseases and possible interventional targets. I recommend this manuscript for publication in current form.

Best regards,

Masaru Tanaka, M.D., Ph.D.